# Predictors of Loss of Functional Independence in Parkinson’s Disease: Results from the COPPADIS Cohort at 2-Year Follow-Up and Comparison with a Control Group

**DOI:** 10.3390/diagnostics11101801

**Published:** 2021-09-29

**Authors:** Diego Santos García, Teresa de Deus Fonticoba, Carlos Cores Bartolomé, Lucía Naya Ríos, Lucía García Roca, Cristina Martínez Miró, Hector Canfield, Silvia Jesús, Miquel Aguilar, Pau Pastor, Marina Cosgaya, Juan García Caldentey, Nuria Caballol, Inés Legarda, Jorge Hernández Vara, Iria Cabo, Lydia López Manzanares, Isabel González Aramburu, María A. Ávila Rivera, Víctor Gómez Mayordomo, Víctor Nogueira, Víctor Puente, Julio Dotor, Carmen Borrué, Berta Solano Vila, María Álvarez Sauco, Lydia Vela, Sonia Escalante, Esther Cubo, Francisco Carrillo Padilla, Juan C. Martínez Castrillo, Pilar Sánchez Alonso, Maria G. Alonso Losada, Nuria López Ariztegui, Itziar Gastón, Jaime Kulisevsky, Marta Blázquez Estrada, Manuel Seijo, Javier Rúiz Martínez, Caridad Valero, Mónica Kurtis, Oriol de Fábregues, Jessica González Ardura, Ruben Alonso Redondo, Carlos Ordás, Luis M. López Díaz, Darrian McAfee, Pablo Martinez-Martin, Pablo Mir

**Affiliations:** 1CHUAC, Complejo Hospitalario Universitario de A Coruña, 15006 A Coruña, Spain; Carlos.Cores.Bartolome@sergas.es (C.C.B.); lucia.naya.rios@gmail.com (L.N.R.); lucia.garcia.toca@gmail.com (L.G.R.); Cristina.Martinez.Miro@sergas.es (C.M.M.); canfield.hector@gmail.com (H.C.); 2CHUF, Complejo Hospitalario Universitario de Ferrol, 15405 A Coruña, Spain; terecorreomovil@gmail.com; 3Unidad de Trastornos del Movimiento, Servicio de Neurología y Neurofisiología Clínica, Instituto de Biomedicina de Sevilla, Hospital Universitario Virgen del Rocío/CSIC/Universidad de Sevilla, 41009 Seville, Spain; smaestre-ibis@us.es (S.J.); pmir@us.es (P.M.); 4CIBERNED (Centro de Investigación Biomédica en Red Enfermedades Neurodegenerativas), 28031 Madrid, Spain; isagaramburu@gmail.com (I.G.A.); jkulisevsky@santpau.cat (J.K.); pmm650@hotmail.com (P.M.-M.); 5Hospital Universitari Mutua de Terrassa, Terrassa, 08221 Barcelona, Spain; miquelaguilar@gmail.com (M.A.); paupastor@mutuaterrassa.es (P.P.); 6Hospital Clínic de Barcelona, 08036 Barcelona, Spain; marinacosgaya@gmail.com; 7Centro Neurológico Oms 42, 07003 Palma de Mallorca, Spain; juangcaldentey@hotmail.com; 8Consorci Sanitari Integral, Hospital Moisés Broggi, Sant Joan Despí, 08970 Barcelona, Spain; nuriacaballol@hotmail.com; 9Hospital Universitario Son Espases, 07120 Palma de Mallorca, Spain; ines.legarda@ssib.es; 10Hospital Universitario Vall d’Hebron, 08035 Barcelona, Spain; hernandezvarajorge76@gmail.com (J.H.V.); odefabregues@gmail.com (O.d.F.); 11Complejo Hospitalario Universitario de Pontevedra (CHOP), 36071 Pontevedra, Spain; icabol@yahoo.es (I.C.); manuel.seijo.martinez@sergas.es (M.S.); 12Hospital Universitario La Princesa, 28006 Madrid, Spain; lydialopez@hotmail.com; 13Hospital Universitario Marqués de Valdecilla, 39011 Santander, Spain; 14Consorci Sanitari Integral, Hospital General de L’Hospitalet, L’Hospitalet de Llobregat, 08906 Barcelona, Spain; asuncion.avila@sanitatintegral.org; 15Hospital Universitario Clínico San Carlos, 28040 Madrid, Spain; vicmayordomo@gmail.com; 16Hospital Da Costa, Burela, 27880 Lugo, Spain; victor.nogueira.fernandez@sergas.es; 17Hospital del Mar, 08003 Barcelona, Spain; Vpuente@parcdesalutmar.cat; 18Hospital Universitario Virgen Macarena, 41009 Sevilla, Spain; juliodotor@gmail.com; 19Hospital Infanta Sofía, 28702 Madrid, Spain; carmenborrue@hotmail.com; 20Institut d’Assistència Sanitària (IAS)—Institut Català de la Salut, 17190 Girona, Spain; berta_solano@hotmail.com; 21Hospital General Universitario de Elche, 03203 Elche, Spain; mariaalsa@hotmail.com; 22Fundación Hospital de Alcorcón, 28922 Madrid, Spain; lvela@fhalcorcon.es; 23Hospital de Tortosa Verge de la Cinta (HTVC), 43500 Tarragona, Spain; sescalant@yahoo.es; 24Complejo Asistencial Universitario de Burgos, 09006 Burgos, Spain; esthercubo@gmail.com; 25Hospital Universitario de Canarias, 38320 San Cristóbal de la Laguna, Spain; fcarpad@gobiernodecanarias.org; 26Hospital Universitario Ramón y Cajal, IRYCIS, 28034 Madrid, Spain; jcmcastrillo@gmail.com; 27Hospital Universitario Puerta de Hierro, 28222 Madrid, Spain; PISANCHEZAL@GMAIL.COM; 28Hospital Álvaro Cunqueiro, Complejo Hospitalario Universitario de Vigo (CHUVI), 36213 Vigo, Spain; gemavarita@gmail.com; 29Complejo Hospitalario de Toledo, 45071 Toledo, Spain; nlariztegui@gmail.com; 30Complejo Hospitalario de Navarra, 31008 Pamplona, Spain; itziar.gaston.zubimendi@cfnavarra.es; 31Hospital de Sant Pau, 08041 Barcelona, Spain; 32Hospital Universitario Central de Asturias, 33011 Oviedo, Spain; marta.blazquez.estrada@gmail.com; 33Hospital Universitario Donostia, 20009 San Sebastián, Spain; JAVIER.RUIZMARTINEZ@osakidetza.eus; 34Hospital Arnau de Vilanova, 46015 Valencia, Spain; carivalero@icloud.com; 35Hospital Ruber Internacional, 28034 Madrid, Spain; mkurtis@ruberinternacional.es; 36Hospital de Cabueñes, 33394 Gijón, Spain; jessardura@yahoo.es; 37Universitario Lucus Augusti (HULA), 27002 Lugo, Spain; alonso_1408@hotmail.es; 38Hospital Rey Juan Carlos, 28933 Madrid, Spain; carlos.ordas@quironsalud.es; 39Complejo Hospitalario Universitario de Orense (CHUO), 32005 Orense, Spain; Luis.Manuel.Lopez.Diaz@sergas.es; 40Laboratory for Cognition and Neural Stimulation, Univeristy of Pennsylvania, Philadelphia, PA 19104, USA; mcafeed@sas.upenn.edu

**Keywords:** activities of daily living, dependency, disability, gait, Parkinson’s disease

## Abstract

Background and objective: The aim of this study was to compare the progression of independence in activities of daily living (ADL) in Parkinson’s disease (PD) patients versus a control group, as well as to identify predictors of disability progression and functional dependency (FD). Patients and Methods: PD patients and control subjects, who were recruited from 35 centers of Spain from the COPPADIS cohort between January 2016 and November 2017 (V0), were included. Patients and subjects were then evaluated again at the 2-year follow-up (V2). Disability was assessed with the Schwab & England Activities of Daily Living Scale (S&E-ADLS) at V0 and V2. FD was defined as an S&E-ADLS score less than 80%. Results: In the PD group, a significant decrease in the S&E-ADLS score from V0 to V2 (N = 507; from 88.58 ± 10.19 to 84.26 ± 13.38; *p* < 0.0001; Cohen’s effect size = −0.519) was observed but not in controls (N = 124; from 98.87 ± 6.52 to 99.52 ± 2.15; *p* = 0.238). When only patients considered functional independent at baseline were included, 55 out of 463 (11.9%) converted to functional dependent at V2. To be a female (OR = 2.908; *p* = 0.009), have longer disease duration (OR = 1.152; *p* = 0.002), have a non-tremoric motor phenotype at baseline (OR = 3.574; *p* = 0.004), have a higher score at baseline in FOGQ (OR = 1.244; *p* < 0.0001) and BDI-II (OR = 1.080; *p* = 0.008), have a lower score at baseline in PD-CRS (OR = 0.963; *p* = 0.008), and have a greater increase in the score from V0 to V2 in UPDRS-IV (OR = 1.168; *p* = 0.0.29), FOGQ (OR = 1.348; *p* < 0.0001) and VAFS-Mental (OR = 1.177; *p* = 0.013) (adjusted R-squared 0.52; Hosmer and Lemeshow test = 0.94) were all found to be independent predictors of FD at V2. Conclusions: In conclusion, autonomy for ADL worsens in PD patients compared to controls. Cognitive impairment, gait problems, fatigue, depressive symptoms, more advanced disease, and a non-tremor phenotype are independent predictors of FD in the short-term.

## 1. Introduction

Parkinson’s disease (PD) is a neurodegenerative disorder in which the symptoms will progress and generate a disability for the patient to carry out his/her activities of daily living (ADL). Loss of functional independence leads to caregiver burden, high resource use, institutionalization, increased risk of death and comorbid complications, worse quality of life (QoL) [1,2], and it is also considered an important outcome of progression [3]. Since there is currently no cure for PD, the goal of treatment in PD is to improve the patient’s symptoms with the intention of increasing his/her autonomy for ADL and achieving a better QoL perception. Knowing what factors will predict when a patient could become functionally dependent is important, since pharmacological and non-pharmacological strategies could be established with the intention of delaying the appearance of functional dependence [4]. In addition, it can also be useful when foreseeing needs such as certain socio-health care and estimating costs both at a particular and global level for a health system [5]. Suggested predictors of functional dependency in PD are older age, higher severity of rigidity and bradykinesia, more severe axial symptoms, dyskinesia, cognitive impairment, and more advanced disease [6,7,8,9]. However, there is a lack of evidence about the progression of disability in PD and the identification of functional dependency predictors in prospective studies. A systematic review about loss of independence in PD published in 2015 included only 14 studies of 15,145 unique references identified [10]. It is noteworthy that many of the studies were carried out many years ago. The authors concluded that little high-quality data on dependency were available and there was heterogeneity in study populations and methodology, showing the necessity for well-conducted studies to better understand the progression of dependency in PD [10]. Specific limitations are the small sample size and the absence of a control group in the majority of the studies. Although many of them have a long-term follow-up, it is key to identify what factors can predict a deterioration in autonomy for ADL in the short-term with the intention to intervene as early as possible.

The aim of this study was to compare the progression of independence in ADL in PD patients vs. a control group from the Spanish cohort COPPADIS (COhort of Patients with PArkinson’s DIsease in Spain), as well as to identify predictors of disability progression and functional dependency in the PD group.

## 2. Methods

PD patients and control subjects, who were recruited from January 2016 to November 2017 and evaluated again at a 2-year follow-up from 35 centers of Spain from the COPPADIS cohort [11], were included in this study. Methodology about COPPADIS-2015 study can be consulted in https://bmcneurol.biomedcentral.com/articles/10.1186/s12883-016-0548-9 (accessed on 22 February 2016) [12]. This is a longitudinal-prospective, 5-year follow-up study designed to analyze natural progression of PD in which patients diagnosed with PD according to UK PD Brain Bank criteria without dementia were included [11].

Data from two visits were obtained in this study: (1) at baseline (V0); (2) at 2-year ± 30 days follow-up (V2). Disability was assessed with the Schwab & England Activities of Daily Living Scale (S&E-ADLS) [13]. This scale is a method of assessing the capabilities of people suffering from impaired mobility. Originally presented in 1968 at Third Symposium on Parkinson’s Disease, Royal College of Surgeons in Edinburgh, the scale assesses the difficulties patients have completing daily activities or chores. It uses percentages to represent how much effort and dependence on others patients need to complete daily chores. S&E-ADLS is a recommended scale for assessing disability in PD patients by the MDS Task Force [14]. The patient must indicate the degree of autonomy for carrying out ADL (the higher the score, the less disability), being from 0% (bed-ridden, vegetative functions such as swallowing, bladder and bowel functions are not functioning) to 100% (completely independent, essentially normal). Functional dependency was defined as an S&E-ADLS score less than 80% (80% = completely independent; 70% = not completely independent) [2,8,15].

Information on sociodemographic aspects, factors related to PD, comorbidity, and treatment was collected. Moreover, motor status, non-motor symptoms (NMS), and QoL was assessed at V0 and at V2 using different validated scales: Hoenh & Yahr (H&Y); Unified Parkinson’s Disease Rating Scale [UPDRS] part III and part IV; Freezing of Gait Questionnaire [FOGQ]); Parkinson’s Disease Cognitive Rating Scale (PD-CRS); Non-Motor Symptoms Scale (NMSS); Beck Depression Inventory-II (BDI-II); Parkinson’s Disease Sleep Scale (PDSS); Neuropsychiatric Inventory (NPI); Questionnaire for Impulsive-Compulsive Disorders in Parkinson’s Disease-Rating Scale (QUIP-RS); Visual Analog Scale-Pain (VAS-Pain); Visual Analog Fatigue Scale (VAFS]); the 39-item Parkinson’s disease Questionnaire (PDQ-39); PQ-10; and the European Health Interview Survey-Quality of Life 8 item index (EUROHIS-QOL8) [12]. In patients with motor fluctuations, the motor assessment was made during the OFF state (without medication in the last 12 h) and during the ON state. On the other hand, the assessment was only performed without medication in patients without motor fluctuations. The same evaluation as for the patients, except for the motor assessment, was conducted in control subjects at V0 and at V2.

### 2.1. Data Analysis

Data were processed using SPSS 20.0 for Windows. For comparisons between groups, the Student’s *t*-test, Mann–Whitney–Wilcoxon test, Chi-square test, or Fisher test were used as appropriate (distribution for variables was verified by one-sample Kolmogorov-Smirnov test). General linear model (GLM) repeated measure was used to test whether the mean differences of the S&E-ADLS score between the two visits (V0 and V2) were significant in both PD patients and controls. Age, gender, and LEDD (levodopa equivalent daily dose) [16] at V0 and at V2 were included as covariates. This test was also applied for testing the difference between V0 and V2 in other variables, being furthermore the S&E-ADLS score at baseline and the change in the S&E-ADLS score from V0 to V2 included as covariates. Cohen’s d formula was applied for measuring the effect size in PD patients. It was considered: small effect = 0.2; medium effect = 0.5; and large effect = 0.8. The Bonferroni method was used as a post-hoc test after ANOVA.

Spearman’s or Pearson’s correlation coefficient, as appropriate, were used for analyzing the relationship between the change from V0 to V2 in the S&E-ADLS score and the change from V0 to V2 in the score of other scales. Correlations were considered weak for coefficient values ≤ 0.29, moderate for values between 0.30 and 0.59, and strong for values ≥ 0.60. The *p*-value was considered significant when it was <0.05.

Linear regression models were used for determining predictive factors of increase disability (i.e., S&E-ADLS score from V0 to V2 decrease; S&E-ADLS score change from V0 to V2 as dependent variable). Any variable with univariate associations with *p*-values < 0.20 were included in a multivariable model, and a backwards selection process was used to remove variables individually until all remaining variables were significant at the 0.05 level [17]. The variables considered for the analysis were: (1) at V0: age, gender, disease duration, total number of non-antiparkinsonian drugs, LEDD, motor phenotype [18], UPDRS-III-OFF, UPDRS-IV, FOGQ, PD-CRS, NMSS, BDI-II, PDSS, NPI, QUIP-RS, VAS-PAIN, VASF-Physical, and VASF-Mental; (2) at V2: to be receiving L-dopa, a MAO-B (monoamine oxidase type B) inhibitor, a COMT (catechol-o-methyl transferase) inhibitor, a dopamine agonist, to practice regular exercise, and to assist regularly to a PD Association; (3) from V0 to V2: the change in total number of non-antiparkinsonian drugs, LEDD, UPDRS-III-OFF, UPDRS-IV, FOGQ, PD-CRS, BDI-II, PDSS, NPI, QUIP-RS, VAS-PAIN, VASF-Physical, and VASF-Mental. All models were adjusted by S&E-ADLS score at V0. Tolerance and variance inflation factor (VIF) were used to detect multicollinearity (multicollinearity was considered problematic when tolerance was less than 0.2 and, simultaneously, the value of VIF 10 and above). By the other hand, binary regression models were used for determining predictive factors of functional dependency at V2 (S&E-ADLS score at V2 < 80% as dependent variable). The variables considered for the analysis were the same as in previous linear models.

### 2.2. Standard Protocol Approvals, Registrations, and Patient Consents

Approval from the *Comité de Ética de la Investigación Clínica de Galicia* from Spain (2014/534; 02/DEC/2014) was obtained. A written informed consent from all participants was signed. COPPADIS-2015 was classified by the AEMPS (*Agencia Española del Medicamento y Productos Sanitarios*) as a Post-authorization Prospective Follow-up study with the code COH-PAK-2014-01.

### 2.3. Data Availability

The data that support the findings of this study are available from the corresponding author upon reasonable request. No computer coding was used in the completion of the current manuscript.

## 3. Results

A total of 507 PD patients (62.63 ± 8.52 years old; 58.8% males; mean disease duration 5.5 ± 4.37 years) and 124 controls (62.59 ± 7.22 years old; 50% males) from the COPPADIS cohort were considered valid for analysis in this study. At V0, the ADLS score was lower in the PD group compared to the control group (88.58 ± 10.19 vs. 98.87 ± 6.52; *p* < 0.0001) (Figure 1). Forty-four out of 507 PD patients (8.7%) presented at V0 functional dependency compared to only one control subject (0.8%) (*p* < 0.0001) (Figure 2A). More than 90% of the controls at V0 and V2 presented a score of 100% in the S&E-ADLS, whereas, in PD patients, a score of 90% was the most frequent (Figure 2B).

In the PD group, a significant decrease in the S&E-ADLS score from V0 to V2 (from 88.58 ± 10.19 to 84.26 ± 13.38; *p* < 0.0001; Cohen’s effect size = −0.519) was observed but not in controls (from 98.87 ± 6.52 to 99.52 ± 2.15; *p* = 0.238) (difference between groups, *p* < 0.0001) (Appendix A and Figure 1). In the PD group, the frequency of functional dependency at V2 was double compared to at V0 (16.8% vs. 8.7%; *p* < 0.0001). No one in the control group presented functional dependency at V2. After adjustment to S&E-ADLS score at V0 and change from V0 to V2 in the S&E-ADLS score, and also age, gender, and LEDD at V0 and V2, different variables changed significantly, indicating a worsening from V0 to V2 in the PD group: Hoehn & Yahr (OFF), UPDRS-III (OFF), UPDRS-IV, FOGQ, number of non-antiparkinsonian drugs, PD-CRS, NMSS, VAS-PAIN, VASF–physical, PDQ-39SI, and PQ-10 (Appendix A). However, the change was significant only for the number of non-antiparkinsonian drugs in the control group. Despite of these differences, a significant difference was not observed between both groups, PD patients and controls, except for the change from V0 to V2 in the S&E-ADLS-score.

A moderate correlation was observed between the change in the S&E-ADLS score from V0 to V2 and the change from V0 to V2 in the PDQ-39SI (r = −0.407; *p* < 0.0001) and the FOGQ (r = −0.328; *p* < 0.0001) (Table 1). Other significant correlations are shown in Table 1. When two groups of PD patients at V2 were considered, patients with functional dependency (N = 85) vs. those with functional independence (N = 422), having functional dependency at V2 was associated with being older, gender (to be a female), and a worse motor and non-motor status at V0, as well as a greater worsening in motor and different NMS from V0 to V2 (Table 2).

In the multivariate analysis and after adjustment to the S&E-ADLS total score, a greater increase of disability (larger decrease in the S&E-ADLs score from V0 to V2) was associated with longer disease duration (*p* = 0.018), a higher total score at V0 in UPDRS-IV (*p* < 0.0001), NPI (*p* = 0.035), and VASF-Physical (*p* = 0.008), a lower score at V0 in PD-CRS (*p* < 0.0001), not practicing regular exercise at V2 (*p* = 0.002), and a greater increase from V0 to V2 in the total score of UPDRS-III (*p* = 0.005), UPDRS-IV (*p* = 0.001), FOGQ (*p* < 0.0001), and VASF-Physical (*p* = 0.004) (Table 3). Change from V0 to V2 in FOGQ (β = −0.316; 95%CI, −1.175–−0.671; *p* < 0.0001) and UPDRS-IV total score at baseline (β = −0.297; 95%CI, −2.023–−0.929; *p* < 0.0001) provided the highest contribution to the model (adjusted R-squared 0.47; Durbin–Watson test = 1.87) (Table 3). In the final model, tolerance was from 0.51 to 0.91 and VIF from 1.07 to 1.98.

In the binary regression model, longer disease duration (OR = 1.098; 95% CI, 1.008–1.195; *p* = 0.031), to be a female (OR = 2.334; 95%CI, 1.127–4.833; *p* = 0.022), to have a higher total score at V0 in FOGQ (OR = 1.199; 95%CI, 1.092–1.317; *p* < 0.0001) and NPI (OR = 1.048; 95%CI, 1.007–1.091; *p* = 0.0.20), to have a lower score in PD-CRS (OR = 0.964; 95%CI, 0.942–0.987; *p* = 0.003), and to have a greater increase in the total score from V0 to V2 in FOGQ (OR = 1.305; 95%CI, 1.195–1.424; *p* < 0.0001) and VAFS-Mental (OR = 1.236; 95%CI, 1.093–1.397; *p* = 0.001) were independently associated with functional dependency at V2, after adjustment to disability (S&E-ADLS at baseline; OR = 0.926; 95%CI, 0.889–0.965; *p* < 0.0001) (adjusted R-squared 0.57; Hosmer and Lemeshow test = 0.27) (Table 4). When only patients considered functionally independent at baseline were included, 55 out of 463 (11.9%) converted to functionally dependent at V2, and similar results were obtained. To be a female (OR = 2.908; 95%CI, 1.312–6.445; *p* = 0.009), to have a longer disease duration (OR = 1.152; 95%CI, 1.052–1.261; *p* = 0.002), to have a non-tremoric motor phenotype at baseline (OR = 3.574; 95%CI, 1.493–8.555; *p* = 0.004), to have a higher score at baseline in FOGQ (OR = 1.244; 95%CI, 1.126–1.374; *p* < 0.0001) and BDI-II (OR = 1.080; 95%CI, 1.021–1.143; *p* = 0.008), to have a lower score in PD-CRS (OR = 0.963; 95%CI, 0.938–0.990; *p* = 0.008), and to have a greater increase in the score from V0 to V2 in UPDRS-IV (OR = 1.168; 95%CI, 1.016–1.344; *p* = 0.0.29), FOGQ (OR = 1.348; 95%CI, 1.221–1.488; *p* < 0.0001) and VAFS-Mental (OR = 1.177; 95%CI, 1.035–1.338; *p* = 0.013) (adjusted R-squared 0.52; Hosmer and Lemeshow test = 0.94) appeared as factors independently associated with functional dependency at V2. In the same model, when variables were included individually as dichotomous variables, the next results were obtained (all models significant): disease duration at V0 ≥ 10 years, OR = 3.983 (95%CI, 1.547–10.252; *p* = 0.004); FOGQ total score at V0 > 5, OR = 5.495 (95%CI, 2.300–13.126; *p* < 0.0001); PD-CRS total score at V0 ≤ 81: OR = 3.788 (95%CI, 95%CI 1.590–9.026; *p* = 0.003); BDI-II total score at V0 > 15: OR = 2.995 (95%CI, 1.168–7.681; *p* = 0.022); FOGQ total score increase from V0 to V2 > 3: OR = 5.913 (95%CI, 2.471–14.147; *p* < 0.0001); and VAFS-Mental total score increase from V0 to V2 > 3: OR = 3.560 (95%CI, 1.207–10.504; *p* = 0.021) (Figure 3). Significant OR for UPDRS-IV total score increase from V0 to V2 was not observed when cut-points were defined (increase in > 2, 3, 4 or 5 points). Finally, when the presence of FOG at baseline (score ≥1 in the item 3 of the FOGQ [19,20]) was considered, experiencing FOG at V0 was an independent predictor of functional dependency at the 2-year follow-up: univariate analysis, OR = 2.945 (95%CI, 1.660–5.223; *p* < 0.0001); multivariate analysis, OR = 2.936 (95%CI, 1.300–6.633; *p* = 0.010; adjusted R-squared 0.47; Hosmer and Lemeshow test = 0.79).

## 4. Discussion

In this study, we observed a worsening in functional independence for ADL in a cohort of 507 PD patients compared to a control group after a 2-year follow-up, being double those PD patients with functional dependency after follow-up compared to at baseline (17% vs. 9%). Moreover, longer disease duration, gender (female), non-tremoric motor phenotype, a worse motor and non-motor status at baseline related to cognitive impairment (PD-CRS), neuropsychiatric symptoms (NPI, BDI-II) and gait problems and motor complications (FOGQ, UPDRS-IV), and a greater impairment in gait problems (FOGQ) and fatigue (VAFS) after 2-year follow-up were identified as predictors of disability/functional dependency in this cohort.

The frequency of patients with functional dependency in our cohort seems to be in line with other studies. The proportion of patients with dependency in inception studies varies between 10–25% at 5 years and 20–50% at 10 years [10]. However, some cohorts reported “dependency or mortality”, being about 15–40% at 5 years and 35–70% at 10 years [10,21,22,23,24,25]. We used the S&E-ADLS for assessing disability and defining functional dependency [13]. Many disability measures are available for application in PD, and the S&E-ADLS is one of the nine recommended scales by the International Parkinson and Movement Disorder Society Task Force [14]. The definition of functional dependency as a S&E-ADLS score <80% was considered according to previous studies [2,8,10]. In a prospective, community-based incident cohort of PD from the Parkinsonism Incidence in North-East Scotland (PINE) study, Macleod et al. observed in 198 patients with PD a rate of development of functional dependency (S&E-ADLS score < 80%) of 14 per 100 person years of follow-up [8]. We observed functional dependency in 9% of 507 PD patients with a mean disease duration of 5 years and a half, being double after a 2-year follow-up. Only one PD patient from our cohort was dead. However, from the initial cohort (N = 689) [11,15], the ADLS score was not recorded for 144 patients (21%). Compared to other studies, mean age and mean disease duration in our cohort were lower and the follow-up period was shorter [8,10,26,27]. It is likely that selection biases and methodological differences between studies can explain the variation in the rates of functional dependency rather than true population differences in dependency risk. Despite the limitations, our study shows that, in a short time (2 years), there is a global worsening in the autonomy of the patients to carry out their ADL, being 12% of functional independent patients at V0 dependent at V2. This does not happen in controls. A few previous studies included a control group [28,29], showing that PD constitutes a significant factor of dependency even in newly diagnosed subjects and also in elderly subjects living at home, as well as that institutionalization occurs more frequently in PD patients than in the general population. Remarkably, motor and NMS in our cohort progressed after the 2-year follow-up, but GLM repeated measures adjusted to disability at baseline and change in disability after the follow-up period showed not significant differences between patients and controls in any variable (but it was observed without this adjustment in many of them [30]), indicating the important influence of motor and NMS progression over functional capacity for ADL. In this context, it is crucial to identify predictive factors of functional dependence in the short term in PD patients because some early interventions could be applied.

Some studies have identified different predictors for increased dependency in PD: older age, age > 60 years old at initiation of levodopa therapy, male sex, greater smoking story, PIGD (postural instability gait difficulty) motor phenotype, more severe axial symptoms, higher severity of rigidity and bradykinesia rather than tremor, no response to levodopa at one year, dyskinesia, cognitive impairment, and more advanced disease [6,7,8,9,10,15,24,25,26,27,28,29,31,32]. To our knowledge, our study is the largest study in which more variables were included in the analysis, both at the baseline level and considering their change over time, and which takes into account the degree of autonomy at baseline. Recently, it has been reported that de novo PD patients with PIGD phenotype have greater disability for ADL [33]. Our findings about age, motor phenotype, motor complications, and cognitive impairment as independent predictors of functional dependency are in line with the literature [6,7,8,9,10,15,24,25,26,27,28,29,30,31,32]. With regard to gender and contrary to Guillard et al. study [32], to be a female increased the probability of functional dependency at 2-year follow-up three-fold, but gender was not a predictor when disability (S&E-ADLS score as continuous variable) was considered as the dependent variable. Recently, Sperens et al. [34] reported that 9 of the 12 domains in the ADL taxonomy showed a change over time (up to 8-year follow-up) in 129 PD patients, with worse scores (N = 53) in women in some scores (shopping and cleaning). In general, it seems that, as PD progresses, gender differences emerge, with men exhibiting more severe Parkinsonian motor features and women experiencing more levodopa-induced dyskinesia, fatigue, feelings of nervousness and sadness, constipation, restless legs, and pain [35,36]. Importantly, in our cohort, a greater global NMS burden was an independent predictor of QoL worsening at 2-year follow-up [37] but not of functional dependency or increased disability for ADL. However, mood and fatigue were factors increasing the probability of functional dependency in the short-term. Higher UPDRS-ADL and PDQ-39 mobility scores have been associated with fatigue [38]. Furthermore, depression impacts QoL and contributes to greater disability in PD, so treatment of depression may in fact improve function [39]. Finally, gait problems were a predictor of disability in our cohort. To score >5 points in the FOGQ and to increase >3 points on the score after 2-year follow-up increased the probability of functional dependency five times. In particular, freezing of gait (FOG) was an independent factor associated with functional dependency in the COPPADIS study baseline cross-sectional analysis (N = 689; adjusted R-squared 0.513; *p* = 0.007) [15]. Again, in this 2-year longitudinal follow-up analysis, FOG was a predictor of functional dependency, so suffering from FOG increased the risk of being functional dependent threefold after 2 years. Balance confidence and FOG are associated with the mobility aspect of health-related QoL [40] and therapies and/or strategies designed to improve gait problems and FOG can benefit QoL and autonomy for ADL [41,42]. For example, compensation strategies seem to be an effective and simple treatment for gait impairment in PD patients [43]. Importantly, other strategies for delaying mobility disability is to practice regular physical exercise [44], as our results suggest.

The present study has some limitations. The most important is that information about ADLS after 2-year follow-up was recorded in 507 out of 689 PD patients (74%) and in 124 out of 207 controls (60%). Thirty-eight patients dropped out of the study (1 death; 2 with change in diagnosis; 35 other reasons) at V2 and for 132 a follow-up was not obtained. In 12 patients evaluated at V2, the ADLS was not assessed. However, this is a limitation observed in other prospective studies, with maintenance rates of 89.8% (380/423) [16], 83.6% (117/140) [45], 73.9% (147/199) [46], and 61.9% (707/1142) [47]. Although a bias regarding the underestimation of functional dependency due to the withdrawal of the most affected patients from the study cannot be ruled out, significant differences between both groups, continuing vs. not continuing in the study at 2-year follow-up, in terms of disability and functional dependency at baseline, were not observed (data not shown). Second, for some variables, the information was not collected in all cases. Third, instead of a specific tool for assessing comorbidity, like Charlson Index or others, the total number of non-antiparkinsonian medications was used as a surrogated marker of comorbidity [48]. Fourth, the diagnosis of FOG was subjective, and an insufficient or excessive diagnosis cannot be ruled out [49]. Fifth, our sample was not fully representative of the PD population due to inclusion and exclusion criteria at baseline (i.e., age limit, no dementia, no severe comorbidities, no second line therapies, etc.) [11,12], which leads to an early PD bias in this cohort. Sixth, all scales or questionnaires used for assessing motor and NMS are validated except PQ-10. This is a very simple question about global QoL perception from 0 (worst) to 10 (best) used in previous studies [46]. To use the PQ-10 takes very little time and provides information similar to the EUROHIS-QOL8 total score [48]. Finally, the sample size of the control group was clearly smaller than in patients. On the contrary, the strengths of our study include the large sample size and the extensive clinical and demographic information recorded with many justified variables included in the models.

In conclusion, our study observes that autonomy for ADLS worsens in patients with PD compared to control subjects and identifies different predictors of functional dependency. The neurologist should be alert and think that those patients with PD with cognitive impairment, non-tremor phenotype, gait problems, fatigue, depressive symptoms, and more advanced disease are more likely to be dependent in the short term.

## Figures and Tables

**Figure 1 diagnostics-11-01801-f001:**
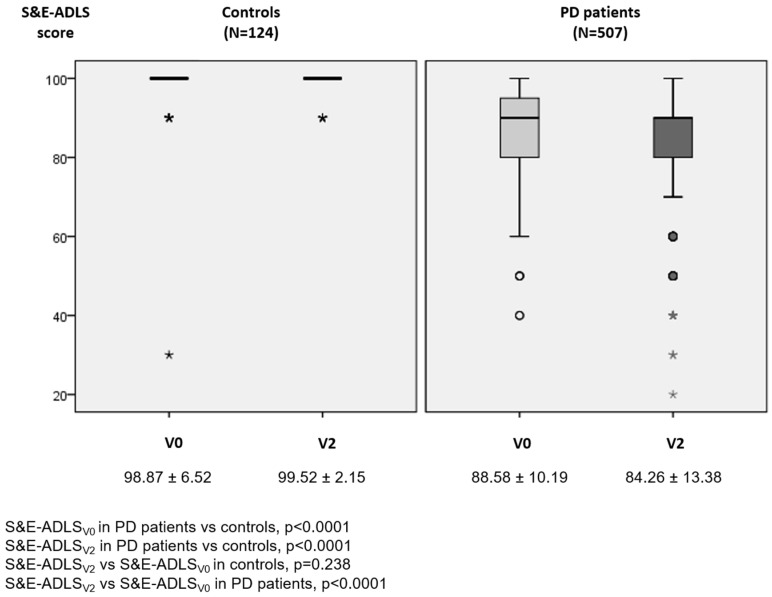
Change in S&E-ADLS score from V0 to V2 in controls (left) and PD patients (right). S&E-ADLS score at V0 in PD vs. controls, * *p* < 0.0001 (Mann–Whitney–Wilcoxon test); S&E-ADLS score at V2 in PD vs. controls, *p* < 0.0001 (Mann–Whitney–Wilcoxon test); S&E-ADLS score at V2 vs. S&E-ADLS score at V0 in controls, *p* = 0.238 (GLM repeated measure adjusted to age and gender); S&E-ADLS score at V2 vs. S&E-ADLS score at V0 in PD patients, *p* < 0.0001 (GLM repeated measure adjusted to age, gender, LEDD at V0 and LEDD at V2). PD, Parkinson’s disease; S&E-ADLS, Schwab & England Activities of Daily Living Scale.

**Figure 2 diagnostics-11-01801-f002:**
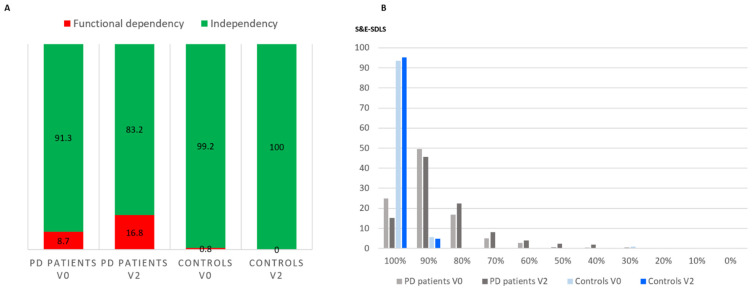
(**A**) Frequency of PD patients and controls presenting functional dependency (S&E-ADLS score < 80%) at V0 and at V2. *p*-values < 0.0001 for functional dependency in PD patients at V2 vs. at V0, in PD patients at V0 vs. in controls at V0, and in PD patients at V2 vs. in controls at V0; (**B**) frequency of PD patients (N = 507; in grey) and controls (N = 124; in blue) presenting different scores in the S&E-ADLS at V0 (baseline) and at V2 (2 years ± 30 days). PD, Parkinson’s disease; S&E-ADLS, Schwab & England Activities of Daily Living Scale.

**Figure 3 diagnostics-11-01801-f003:**
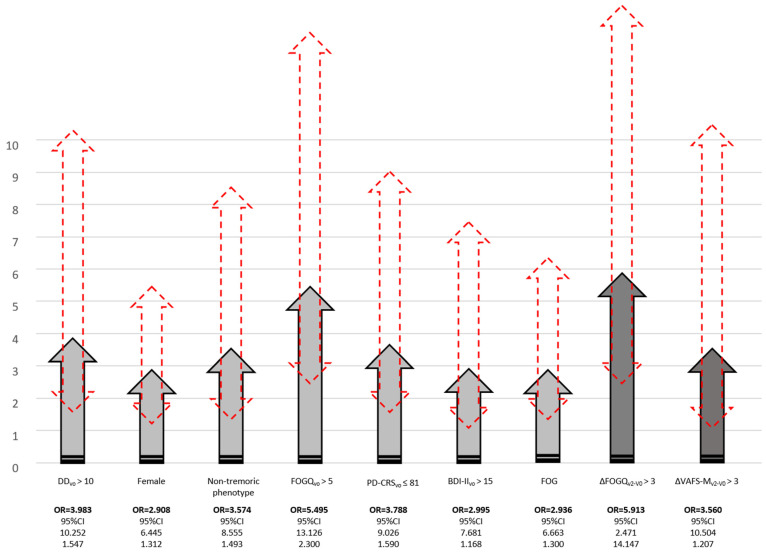
Predictors of functional disability (S&E-ADLS score < 80%) at V2. Light grey arrows show the OR of variables from baseline visit (V0). Dark grey arrows show the OR of change of variables from V0 to V2. Red arrows with discontinue line represent the 95%CI. BDI-II, Beck Depression Inventory-II; DD, disease duration; FOGQ, Freezing Of Gait Questionnaire; PD-CRS, Parkinson’s Disease Cognitive Rating Scale; VAFS, Visual Analog Fatigue Scale.

**Table 1 diagnostics-11-01801-t001:** Correlations between the change in the S&E-ADLS score and other disease-related variables in PD patients from V0 (baseline) to V2 (2 years ± 1 month).

	S&E-ADLSV2—S&E-ADLSV0	N	*p*
Age at baseline	−0.017	507	0.704
Disease duration (at V0)	−0.023	487	0.609
N. of non-antipark. drugs (at V0)	0.002	507	0.965
**Change at V2 (V2 vs. V0)**			
LEDD			
Number of non-antipark. Drugs	−0.067	483	0.143
UPDRS-III (OFF)	−0.250	441	<0.0001
UPDRS-IV	−0.155	485	0.001
FOGQ	−0.328	502	<0.0001
PD-CRS	0.003	494	0.955
NMSS	−0.243	499	<0.0001
BDI-II	−0.214	500	<0.0001
PDSS	0.168	501	<0.0001
QUIP-RS	−0.033	450	0.482
NPI	−0.236	389	<0.0001
VAS-PAIN	−0.148	497	0.001
VASF–physical	−0.152	497	0.001
VASF–mental	−0.199	497	<0.0001
PDQ-39SI	−0.407	499	<0.0001
PQ-10	0.137	497	0.002
EUROHIS-QOL8	0.193	501	<0.0001

Spearman correlation test were applied. The sample was different for each analysis because the data were not available for all patients for other variables. ADLS, Schwab & England Activities of Daily Living Scale; BDI-II, Beck Depression Inventory-II; DA, dopamine agonist; EUROHIS-QOL8, European Health Interview Survey-Quality of Life 8 item index; FOGQ, Freezing Of Gait Questionnaire; LEDD, levodopa equivalent dauly dose; N, number; NMSS, Non-Motor Symptoms Scale; NPI, Neuropsychiatric Inventory; PD-CRS, Parkinson’s Disease Cognitive Rating Scale; PDQ-39SI, 39-item Parkinson’s disease Quality of Life Questionnaire Summary Index score; PDSS, Parkinson’s Disease Sleep Scale; QUIP-RS, Questionnaire for Impulsive-Compulsive Disorders in Parkinson’s Disease-Rating Scale; TS, total score; UPDRS, Unified Parkinson’s Disease Rating Scale; VAFS, Visual Analog Fatigue Scale; VAS-Pain, Visual Analog Scale-Pain.

**Table 2 diagnostics-11-01801-t002:** Different PD related variables in PD patients at V2 (2 years ± 1 month) with functional dependency (N = 85) vs. those ones with functional independency (N = 422).

	All Sample (N = 507)	Functional Independency (N = 422)	Functional Dependency (N = 85)	*p*
Age at baseline	62.63 ± 8.55	62.47 ± 8.41	63.41 ± 9.04	0.187
Males (%)	58.8	60.7	49.4	0.047
Disease duration (at V0)	5.5 ± 4.21	4.99 ± 3.59	8 ± 5.83	<0.0001
**At V0**				
N. of non-antipark. drugs	2.52 ± 2.38	2.33 ± 2.2	3.44 ± 2.94	0.002
LEDD	577.96 ± 382.76	537.56 ± 386.79	785.85 ± 498.27	<0.0001
UPDRS-III (OFF)	22.56 ± 10.56	20.89 ± 10	27.98 ± 11.13	<0.0001
UPDRS-IV	2 ± 2.41	1.63 ± 2.03	3.87 ± 3.21	<0.0001
FOGQ	3.74 ± 4.67	2.89 ± 3.87	7.87 ± 5.92	<0.0001
Non-tremotic motor phenotype (%)	52.3	47.9	74.1	<0.0001
PD-CRS	91.98 ± 15.75	93.64 ± 15.21	83.71 ± 15.84	<0.0001
NMSS	44.72 ± 37.48	40.24 ± 35.82	66.94 ± 37.82	<0.0001
BDI-II	8.27 ± 6.95	7.38 ± 6.45	12.67 ± 7.64	<0.0001
PDSS	117.16 ± 24.53	118.9 ± 24.3	108.52 ± 23.95	<0.0001
QUIP-RS	4.41 ± 8.65	4.26 ± 8.72	5.13 ± 8.32	0.213
NPI	5.78 ± 7.93	4.63 ± 0.23	11.23 ± 11.93	<0.0001
VAS-PAIN	2.57 ± 2.91	2.34 ± 2.79	3.71 ± 3.25	0.001
VASF–physical	2.58 ± 2.91	2.35 ± 2.79	3.71 ± 3.25	<0.0001
VASF–mental	2.08 ± 2.05	1.93 ± 2.4	2.78 ± 2.88	0.019
PDQ-39SI	16.54 ± 12.9	14.32 ± 11.24	27.52 ± 14.9	<0.0001
PQ-10	7.28 ± 1.53	7.43 ± 1.47	6.54 ± 1.63	<0.0001
EUROHIS-QOL8	3.78 ± 0.53	3.85 ± 0.52	3.44 ± 0.48	<0.0001
S&E-ADLS	88.58 ± 10.19	90.62 ± 8.04	78.47 ± 13.31	<0.0001
**At V2**				
To be receiving levodopa	89.4	87.7	97.6	0.008
To be receiving a dopamine agonist	72.4	72.8	70.7	0.706
To be receiving a MAO-B inhibitor	76.9	77.8	72.3	0.274
To be receiving a COMT inhibitor	28.9	25.1	48.2	<0.0001
To practice regular execise	74.3	76.8	61.9	0.004
To assist regularly to a PD association	16.3	16.2	16.7	0.913
**Change at V2 (V2 vs. V0)**				
LEDD	+186.38 ± 326.51	+175.19 ± 307.72	+242.73 ± 405.89	0.102
Number of non-antipark. Drugs	+0.55 ± 1.56	+0.5. ± 1.53	+0.83 ± 1.67	0.154
UPDRS-III (OFF)	+3.3 ± 10.26	+2.74 ± 3.76	+5.92 ± 12.08	0.01
UPDRS-IV	+0.67 ± 2.52	+0.55 ± 2.19	+1.23 ± 3.71	0.052
FOGQ	+1.17 ± 4.16	+0.78 ± 3.64	+3.11 ± 5.75	<0.0001
PD-CRS	−1.83 ± 11.85	−1.22 ± 10.99	−4.89 ± 15.2	0.03
NMSS	+8.29 ± 34.47	+6.5 ± 31.68	+17.11 ± 45.03	0.015
BDI-II	+0.28 ± 7.72	+0.34 ± 7.6	−0.06 ± 8.34	0.663
PDSS	+0.74 ± 26.09	+0.83 ± 24.88	+0.26 ± 31.59	0.893
QUIP-RS	+0.05 ± 9.14	+0.19 ± 8.28	−0.59 ± 12.51	0.128
NPI	+0.35 ± 8.82	+0.26 ± 7.92	+0.74 ± 12.04	0.325
VAS-PAIN	+0.33 ± 3.33	+0.29 ± 3.22	+0.54 ± 3.8	0.554
VASF–physical	+0.31 ± 2.97	+0.27 ± 2.9	+0.48 ± 3.33	0.367
VASF–mental	+0.11 ± 2.84	−0.03 ± 2.63	+0.78 ± 3.65	0.038
PDQ-39SI	+3.5 ± 12.1	+2.2 ± 10.4	+9.94 ± 16.96	<0.0001
PQ-10	−0.15 ± 1.73	−0.13 ± 1.71	−0.26 ± 1.83	0.351
EUROHIS-QOL8	−0.02 ± 0.59	−0.04 ± 0.57	+0.09 ± 0.67	0.151
S&E-ADLS	−4.32 ± 11.76	−1.49 ± 8.32	−18.35 ± 15.72	<0.0001

Chi-square and Mann–Whitney–Wilcoxon test were used. The data were not available for all cases (N = 507) in all analyses, being the lowest for the change from V0 to V2 in the NPI score (N = 388) and after this the NPI at baseline (N = 428). ADLS, Schwab & England Activities of Daily Living Scale; BDI-II, Beck Depression Inventory-II; DA, dopamine agonist; EUROHIS-QOL8, European Health Interview Survey-Quality of Life 8 item index; FOGQ, Freezing Of Gait Questionnaire; LEDD, levodopa equivalent dauly dose; N, number; NMSS, Non-Motor Symptoms Scale; NPI, Neuropsychiatric Inventory; PD-CRS, Parkinson’s Disease Cognitive Rating Scale; PDQ-39SI, 39-item Parkinson’s disease Quality of Life Questionnaire Summary Index score; PDSS, Parkinson’s Disease Sleep Scale; QUIP-RS, Questionnaire for Impulsive-Compulsive Disorders in Parkinson’s Disease-Rating Scale; TS, total score; UPDRS, Unified Parkinson’s Disease Rating Scale; VAFS, Visual Analog Fatigue Scale; VAS-Pain, Visual Analog Scale-Pain.

**Table 3 diagnostics-11-01801-t003:** Linear regression model about factors associated with disability progression after 2-year follow-up (change in the S&E-ADLS score at V2 compared to at V0).

	βa	βb	95% ICa	95% ICb	pa	pb
At V0 (baseline)						
Age	0.000	N. A.	−0.121–0.120	N. A.	0.996	N. A.
Gender (female)	−0.048	N. A.	−3.243–0.957	N. A.	0.285	N. A.
Disease duration (at V0)	−0.107	−0.121	−0.556–−0.066	−0.548–−0.051	0.013	0.018
**At V0**						
Number of non-antipark. drugs	−0.028	N. A.	−0.571–0.294	N. A.	0.529	N. A.
LEDD	−0.049	N. A.	−0.004–0.001	N. A.	0.275	N. A.
UPDRS-III (OFF)	−0.049	N. A.	−0.159–0.047	N. A.	0.283	N. A.
UPDRS-IV	−0.122	−0.297	−1.008–−0.169	−2.023–−0.929	0.006	<0.0001
FOGQ	−0.052	N. A.	−0.433–0.044	N. A.	0.241	N. A.
Non-tremoric motor phenotype	0.035	N. A.	−1.228–2.884	N. A.	0.429	N. A.
PD-CRS	0.093	0.16	0.004–0.135	0.057–0.180	0.038	<0.0001
NMSS	−0.047	N. A.	−0.042–0.013	N. A.	0.291	N. A.
BDI-II	−0.021	N. A.	−0.183–0.114	N. A.	0.646	N. A.
PDSS	−0.000	N. A.	−0.042–0.042	N. A.	0.997	N. A.
QUIP-RS	−0.014	N. A.	−0.142– 0.104	N. A.	0.759	N. A.
NPI	−0.114	−0.092	−0.315–−0.029	−0.265–−0.010	0.018	0.035
VAS-PAIN	−0.046	N. A.	−0.537–0.169	N. A.	0.307	N. A.
VASF–physical	−0.063	−0.138	−0.656–0.108	−1.054–−0.161	0.159	0.008
VASF–mental	−0.014	N. A.	−0.476–0.349	N. A.	0.762	N. A.
S&E-ADLS	−0.264	−0.533	−0.309–−0.174	−0.713–−0.497	<0.0001	<0.0001
**At V2**						
To be receiving levodopa	−0.050	N. A.	−5.248–−1.420	N. A.	0.26	N. A.
To be receiving a dopamine agonist	0.035	N. A.	−1.391–3.216	N. A.	0.437	N. A.
To be receiving a MAO-B inhibitor	0.057	N. A.	−0.866–4.036	N. A.	0.204	N. A.
To be receiving a COMT inhibitor	−0.059	N. A.	−3.800–0.759	N. A.	0.191	N. A.
To practice regular exercise	0.094	0.128	0.173–4.908	1.313–5.958	0.036	0.002
To assist regularly to a PD association	−0.010	N. A.	−3.131–2.475	N. A.	0.818	N. A.
**Change at V2 (V2 vs. V0**)						
LEDD	−0.102	N. A.	−0.007–−0.001	N. A.	0.025	N. A.
Number of non-antipark. drugs	−0.138	N. A.	−1.707–−0.384	N. A.	0.002	N. A.
UPDRS-III (OFF)	−0.256	−0.122	−0.404–0.192	−0.233–−0.043	<0.0001	0.005
UPDRS-IV	−0.149	−0.154	−1.103–−0.282	−1.123–−0.272	0.001	0.001
FOGQ	−0.405	−0.316	−1.436–−1.015	−1.175–−0.671	<0.0001	<0.0001
PD-CRS	0.067	N. A.	−0.021–0.155	N. A.	0.135	N. A.
NMSS	−0.282	N. A.	−0.125–−0.068	N. A.	<0.0001	N. A.
BDI-II	−0.264	N. A.	−0.397–0.131	N. A.	<0.0001	N. A.
PDSS	0.182	N. A.	0.043–0.121	N. A.	<0.0001	N. A.
QUIP-RS	−0.008	N. A.	−0.133–0.111	N. A.	0.865	N. A.
NPI	−0.168	N. A.	−0.366–−0.095	N. A.	0.001	N. A.
VAS-PAIN	−0.127	N. A.	−0.758–−0.140	N. A.	0.004	N. A.
VASF–physical	−0.116	−0.140	−0.804–−0.112	−0.920–−0.178	0.01	0.004
VASF–mental	−0.193	N. A.	−1.156–−0.441	N. A.	<0.0001	N. A.

Dependent variable: ΔS&E-ADLS = S&E-ADLSV2–S&E-ADLSV0. β standardized coefficient and 95% IC are shown. a, univariate analysis; b, multivariate analysis (Durbin–Watson test = 1.87; R2 = 0.47). BDI-II, Beck Depression Inventory-II; COMT, catechol-o-methyl transferase; DA, dopamine agonist; FOG, freezing of gait; FOGQ, Freezing Of Gait Questionnaire; LEDD, levodopa equivalent daily dose (mg); MAO-B, monoamine oxidase type B; N. A., not applicable; NMS, non-motor symptoms; NMSS, Non-Motor Symptoms Scale; NPI, Neuropsychiatric Inventory; PD, Parkinson’s disease; PD-CRS, Parkinson’s Disease Cognitive Rating Scale; PDQ-39SI, 39-item Parkinson’s Disease Quality of Life Questionnaire Summary Index; PDSS, Parkinson’s Disease Sleep Scale; QoL, Quality of life; QUIP-RS, Questionnaire for Impulsive-Compulsive Disorders in Parkinson’s Disease-Rating Scale; S&E-ADLS, Schwab & England Activities of Daily Living Scale; UPDRS, Unified Parkinson’s Disease Rating Scale; VAFS, Visual Analog Fatigue Scale; VAS-Pain, Visual Analog Scale-Pain.

**Table 4 diagnostics-11-01801-t004:** Factors associated with functional dependency at V2 (S&E-ADLS score < 80%).

	ORa	ORb	95% ICa	95% ICb	pa	pb
At V0 (baseline)						
Age	1.013	N. A.	0.985–1.042	N. A.	0.354	N. A.
Gender (female)	1.602	2.334	1.003–2.559	1.127–4.833	0.049	0.022
Disease duration (at V0)	1.163	1.098	1.001–1.227	1.008–1.195	<0.0001	0.031
**At V0**						
Number of non-antipark. drugs	1.19	N. A.	1.088–1.302	N. A.	<0.0001	N. A.
LEDD	1.001	N. A.	1.001–1.002	N. A.	<0.0001	N. A.
UPDRS-III (OFF)	1.063	N. A.	1.040–1.087	N. A.	<0.0001	N. A.
UPDRS-IV	1.377	N. A.	1.255–1.511	N. A.	<0.0001	N. A.
FOGQ	1.208	1.199	1.152–1.267	1.092–1.317	<0.0001	<0.0001
Non-tremoric motor phenotype (%)	3.119	N. A.	1.851–5.254	N. A.	<0.0001	N. A.
PD-CRS	0.959	0.964	0.944–0.975	0.942–0.987	<0.0001	<0.0001
NMSS	1.016	N. A.	1.010–1.022	N. A.	<0.0001	N. A.
BDI-II	1.101	N. A.	1.066–1.137	N. A.	<0.0001	N. A.
PDSS	0.985	N. A.	0.997–0.993	N. A.	0.001	N. A.
QUIP-RS	1.011	N. A.	0.985–1.037	N. A.	0.409	N. A.
NPI	1.088	1.048	1.057–1.120	1.007–1.091	<0.0001	0.02
VAS-PAIN	1.162	N. A.	1.076–1.254	N. A.	<0.0001	N. A.
VASF–physical	1.268	N. A.	1.162–1.384	N. A.	<0.0001	N. A.
VASF–mental	1.133	N. A.	1.038–1.236	N. A.	0.005	N. A.
S&E-ADLS	0.895	0.926	0.872–0.920	0.889–0.965	<0.0001	<0.0001
**At V2**						
To be receiving levodopa	5.77	N. A.	1.377–24.175	N. A.	0.016	N. A.
To be receiving a dopamine agonist	0.808	N. A.	0.533–1.515	N. A.	0.687	N. A.
To be receiving a MAO-B inhibitor	0.749	N. A.	0.439–1.276	N. A.	0.288	N. A.
To be receiving a COMT inhibitor	2.764	N. A.	1.704–4.485	N. A.	<0.0001	N. A.
To practice regular exercise	0.488	N. A.	0.173–4.908	N. A.	0.005	N. A.
To assist regularly to a PD association	1.024	N. A.	0.545–1.922	N. A.	0.942	N. A.
**Change at V2 (V2 vs. V0)**						
LEDD	1.001	N. A.	1.000–1.001	N. A.	0.093	N. A.
Number of non-antipark. drugs	1.141	N. A.	0.985–1.322	N. A.	0.079	N. A.
UPDRS-III (OFF)	1.03	N. A.	1.006–1.054	N. A.	0.014	N. A.
UPDRS-IV	1.109	N. A.	1.012–1.216	N. A.	0.027	N. A.
FOGQ	1.137	1.305	1.075–1.201	1.195–1.424	<0.0001	<0.0001
PD-CRS	0.974	N. A.	0.954–0.994	N. A.	0.011	N. A.
NMSS	1.009	N. A.	1.002–1.015	N. A.	0.011	N. A.
BDI-II	0.993	N. A.	0.963–1.024	N. A.	0.662	N. A.
PDSS	0.999	N. A.	0.090–1.008	N. A.	0.853	N. A.
QUIP-RS	0.991	N. A.	0.964–1.018	N. A.	0.494	N. A.
NPI	1.006	N. A.	0.978–1.035	N. A.	0.677	N. A.
VAS-PAIN	1.023	N. A.	0.953–1.098	N. A.	0.533	N. A.
VASF–physical	1.025	N. A.	0.947–1.109	N. A.	0.548	N. A.
VASF–mental	1.106	1.236	1.019–1.200	1.093–1.379	0.016	0.001

Dependent variable: Functional dependency (S&E-ADLS < 80% at V2). OR and 95% IC are shown. a, univariate analysis; b, multivariate analysis (Hosmer and Lemeshow test = 0.27; R2 = 0.57). BDI-II, Beck Depression Inventory-II; COMT, catechol-o-methyl transferase; DA, dopamine agonist; FOG, freezing of gait; FOGQ, Freezing Of Gait Questionnaire; LEDD, levodopa equivalent daily dose (mg); MAO-B, monoamine oxidase type B; N. A., not applicable; NMS, non-motor symptoms; NMSS, Non-Motor Symptoms Scale; NPI, Neuropsychiatric Inventory; PD, Parkinson’s disease; PD-CRS, Parkinson’s Disease Cognitive Rating Scale; PDQ-39SI, 39-item Parkinson’s Disease Quality of Life Questionnaire Summary Index; PDSS, Parkinson’s Disease Sleep Scale; QoL, Quality of life; QUIP-RS, Questionnaire for Impulsive-Compulsive Disorders in Parkinson’s Disease-Rating Scale; S&E-ADLS, Schwab & England Activities of Daily Living Scale; UPDRS, Unified Parkinson’s Disease Rating Scale; VAFS, Visual Analog Fatigue Scale; VAS-Pain, Visual Analog Scale-Pain.

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
