# Peer review of "Predictors of Loss of Functional Independence in Parkinson’s Disease: Results from the COPPADIS Cohort at 2-Year Follow-Up and Comparison with a Control Group"

_diagnostics, 2021, doi:10.3390/diagnostics11101801_

Round 1

Reviewer 1 Report

Dear Authors,

I have carefully read your paper, which studies the predictors of loss of functional independence in Parkinson´s Disease.

The article is very interesting and well-written. Is easy to follow authors’ thoughts and reasoning. I only have a few comments that I think should be adress to improve your paper:

Line 141: What do you whant to mean with “PQ-10”?
In tables 1 and 2 - The description of the acronyms for the variables is missing: “PDQ-39SI, PQ-10 (again what do you mean with this?), EUROHIS-QOL8”.

Please confirm that in Table 2 you really mean to write "V0" in row 29 of the first column. In the lines 173-175 you mention this variables only for “V2”.

You mention in line 260: “In the multivariate analysis and after adjustment to the S&E-ADLS total score, a greater increase of disability (larger decrease in the S&E-ADLs score from V0 to V2) was associated with … to practice regular exercise at V2 (p=0.002)”. I was hoping to read a possible justification for this finding in the discussion section since the current scientific evidence points in the exact opposite direction, i.e., that regular physical exercise is a protective and predictive factor for greater functional autonomy in this population.

Author Response

Reviewer 1

I have carefully read your paper, which studies the predictors of loss of functional independence in Parkinson´s Disease.

The article is very interesting and well-written. Is easy to follow authors’ thoughts and reasoning. I only have a few comments that I think should be adress to improve your paper:

Line 141: What do you whant to mean with “PQ-10”?

AUTHORS – Thank you very much for your comment. PQ-10 is a rating of global perceived QoL (PQ-10) on a scale from 0 (worst) to 10 (best) (Santos García D, de la Fuente-Fernández R. Impact of non-motor symptoms on health-related and perceived quality of life in Parkinson's disease. J Neurol Sci 2013;332:136-40). To use the PQ-10 takes very little time and provides information similar to the EUROHIS-QOL8 total score (Santos García et al. Non-motor symptoms burden, mood, and gait problems are the most significant factors contributing to a poor quality of life in non-demented Parkinson's disease patients: Results from the COPPADIS Study Cohort. Parkinsonism Relat Disord 2019;66:151-7). We included a comment in limitations: “Finally, all scales or questionnaires used for assessing motor and NMS are validated ex-cept PQ-10. This is a very simple question about global QoL perception from 0 (worst) to 10 (best) used in previous studies [48]. To use the PQ-10 takes very little time and provides information similar to the EUROHIS-QOL8 total score [50]”. 

 In tables 1 and 2 - The description of the acronyms for the variables is missing: “PDQ-39SI, PQ-10 (again what do you mean with this?), EUROHIS-QOL8”.

AUTHORS – Thank you for your comment. The acronyms has been added. There is no an acronym for PQ-10. Moreover, we added the full term for EUROHIS-QOL8 the first time that it appears in the text: European Health Interview Survey-Quality of Life 8 item index.

Please confirm that in Table 2 you really mean to write "V0" in row 29 of the first column. In the lines 173-175 you mention this variables only for “V2”.

AUTHORS – Thank you very much for your comment and reviewing so carefully the manuscript. You are right. It is V2. The mistake has been corrected.

You mention in line 260: “In the multivariate analysis and after adjustment to the S&E-ADLS total score, a greater increase of disability (larger decrease in the S&E-ADLs score from V0 to V2) was associated with … to practice regular exercise at V2 (p=0.002)”. I was hoping to read a possible justification for this finding in the discussion section since the current scientific evidence points in the exact opposite direction, i.e., that regular physical exercise is a protective and predictive factor for greater functional autonomy in this population.

AUTHORS – Thank you very much for your important comment. There is a mistake. In this case, the value of β is positive, so to practice exercise is associated with an increase in ADLS score (dependent variable) and viceversa. In fact, 76.8% of the patients with functional independency practiced regular exercise compared to 61.9% of them with functional dependency (table 2, p=0.004). It has been corrected as: “not practicing regular exercise”. Thanks again.

  Reviewer 1

I have carefully read your paper, which studies the predictors of loss of functional independence in Parkinson´s Disease.

The article is very interesting and well-written. Is easy to follow authors’ thoughts and reasoning. I only have a few comments that I think should be adress to improve your paper:

Line 141: What do you whant to mean with “PQ-10”?

AUTHORS – Thank you very much for your comment. PQ-10 is a rating of global perceived QoL (PQ-10) on a scale from 0 (worst) to 10 (best) (Santos García D, de la Fuente-Fernández R. Impact of non-motor symptoms on health-related and perceived quality of life in Parkinson's disease. J Neurol Sci 2013;332:136-40). To use the PQ-10 takes very little time and provides information similar to the EUROHIS-QOL8 total score (Santos García et al. Non-motor symptoms burden, mood, and gait problems are the most significant factors contributing to a poor quality of life in non-demented Parkinson's disease patients: Results from the COPPADIS Study Cohort. Parkinsonism Relat Disord 2019;66:151-7). We included a comment in limitations: “Finally, all scales or questionnaires used for assessing motor and NMS are validated ex-cept PQ-10. This is a very simple question about global QoL perception from 0 (worst) to 10 (best) used in previous studies [48]. To use the PQ-10 takes very little time and provides information similar to the EUROHIS-QOL8 total score [50]”. 

 In tables 1 and 2 - The description of the acronyms for the variables is missing: “PDQ-39SI, PQ-10 (again what do you mean with this?), EUROHIS-QOL8”.

AUTHORS – Thank you for your comment. The acronyms has been added. There is no an acronym for PQ-10. Moreover, we added the full term for EUROHIS-QOL8 the first time that it appears in the text: European Health Interview Survey-Quality of Life 8 item index.

Please confirm that in Table 2 you really mean to write "V0" in row 29 of the first column. In the lines 173-175 you mention this variables only for “V2”.

AUTHORS – Thank you very much for your comment and reviewing so carefully the manuscript. You are right. It is V2. The mistake has been corrected.

You mention in line 260: “In the multivariate analysis and after adjustment to the S&E-ADLS total score, a greater increase of disability (larger decrease in the S&E-ADLs score from V0 to V2) was associated with … to practice regular exercise at V2 (p=0.002)”. I was hoping to read a possible justification for this finding in the discussion section since the current scientific evidence points in the exact opposite direction, i.e., that regular physical exercise is a protective and predictive factor for greater functional autonomy in this population.

AUTHORS – Thank you very much for your important comment. There is a mistake. In this case, the value of β is positive, so to practice exercise is associated with an increase in ADLS score (dependent variable) and viceversa. In fact, 76.8% of the patients with functional independency practiced regular exercise compared to 61.9% of them with functional dependency (table 2, p=0.004). It has been corrected as: “not practicing regular exercise”. Thanks again.

Reviewer 2 Report

Summary: In the present study, the authors have sought to identify the factors that are predictive of disability progression and functional dependency in Parkinson’s patients compared to controls. They have found that the independence of an individual to conduct activities of daily life deteriorates significantly more in PD patients compared controls.

Introduction is good. I think the introduction section is well-written, methods section has good details, results and discussion sections are also nicely done. I have some major and minor concerns that I feel should be addressed before recommending for publication.

Major:

  • Why is there a huge sample difference between the PD patients and controls? It is a big difference and could affect the outcome of the analysis.
  • Do these predictive factors change in the same individual simultaneously?
  • I am not sure how early interventions could be applied to information generated by this study. These predictive factors might be a result of the disease and addressing them pharmacologically won’t help with the progression of the disease and pharmacological intervention to relive these symptoms has been going on for a very long time.
  • What kind of interventions are the authors proposing for gender differences as a factor?

Minor:

  • COPPADIS – full form?
  • English needs to be improved a little.
  • Line 16 change pacient to patient.
  • Line 115 – 5-year or 2-year?

Author Response

Reviewer 2

Summary: In the present study, the authors have sought to identify the factors that are predictive of disability progression and functional dependency in Parkinson’s patients compared to controls. They have found that the independence of an individual to conduct activities of daily life deteriorates significantly more in PD patients compared controls.

Introduction is good. I think the introduction section is well-written, methods section has good details, results and discussion sections are also nicely done. I have some major and minor concerns that I feel should be addressed before recommending for publication.

Major:

Why is there a huge sample difference between the PD patients and controls? It is a big difference and could affect the outcome of the analysis.

AUTHORS – Thank you very much for your comment. We agree with you that this is a limitation of the study. In the protocol of COPPADIS study a sample of 400 controls was planned. However, finally only 207 controls were included (Santos García et al. Eur J Neurol 2019;26:1399-407). Despite of this limitation it seems clear that ADLS didn´t change in controls (from 98.87 ± 6.52 to 99.52 ± 2.15; p=0.238) but it did in patients with a Cohen´s effect size of -0.519. We added a sentence in limitations: “The sample size of the control group was clearly smaller than in patients”.

Do these predictive factors change in the same individual simultaneously?

I am not sure how early interventions could be applied to information generated by this study. These predictive factors might be a result of the disease and addressing them pharmacologically won’t help with the progression of the disease and pharmacological intervention to relive these symptoms has been going on for a very long time. What kind of interventions are the authors proposing for gender differences as a factor?

AUTHORS – Thank you very much for your comment. It is clear that there are variables that we cannot modify, such as gender or disease duration. However, we can at least try it on others, such as mood, cognition, gait disturbances or fatigue, with drugs or interventions. For example, compensation strategies seem to be an effective and simple treatment for gait impairment in PD patients (Tosserans et al. Neurology 2021). Importantly, other strategies for delaying mobility disability is to practice regular physical exercise (King et al. Phys Ther 2009;89:384-93) or to treat depression. We added a comment about this in Discussion.

 Minor:

COPPADIS – full form?

AUTHORS – Thank you very much for your comment.  We included the full form the first time that the term COPPADIS appear: COPPADIS (COhort of Patients with PArkinson's DIsease in Spain).

 English needs to be improved a little.

AUTHORS – Thank you for your comment. English style has been reviewed by Darrian McAfee, English native speaker from Univeristy of Maryland School of Medicine, USA.

Line 16 change pacient to patient.

AUTHORS – Thank you for your comment. The typo has been corrected.

Line 115 – 5-year or 2-year?

AUTHORS – Thank you for your comment. The results of this study is about 2-year follow-up. However, the planned follow-up is 5 years. In this sentence we inform about the COPPADIS study (reference 11: https://bmcneurol.biomedcentral.com/articles/10.1186/s12883-016-0548-9). Scientific production of this project can be consulted in https://curemoselparkinson.org/proyectos-en-desarrollo/coppadis/.

Finally, the terms “At V0”, “At V2” and “Change from V0 to V2 (V2-V0)” in tables appear now in bold to highlight the fields. Please, the green part of the figures is an artifact and should not appear (it should be all white as the background)

Round 2

Reviewer 2 Report

Thanks to the authors for editing the manuscript and addressing my concerns. I find the edits satisfactory and recommend it for publication.